# Synthesis and Characterization of Imidazolium-Based Ionenes

**DOI:** 10.3390/molecules30193961

**Published:** 2025-10-02

**Authors:** Eveline Elisabeth Kanatschnig, Florian Wanghofer, Markus Wolfahrt, Sandra Schlögl

**Affiliations:** 1Polymer Competence Center Leoben GmbH, Sauraugasse 1, 8700 Leoben, Austriamarkus.wolfahrt@pccl.at (M.W.); 2Institute of Chemistry of Polymeric Materials, Technical University of Leoben, Otto Glöckel-Strasse 2, 8700 Leoben, Austria

**Keywords:** ionic liquids, ionenes, polymerization, thermoplastics, characterization

## Abstract

Owing to multiple non-covalent interactions, ionic groups impart unique chemical and physical properties into polymers including ion conductivity/mobility, permeation, and intrinsic healability. Ionenes contain ionic groups in their polymer backbone, which offer great versatility in polymer design. Herein, selected aliphatic and aromatic imidazoles were synthesized, which were used as monomeric building blocks for the preparation of thermoplastic ionenes by following a S*n*2 step growth reaction across organic halides. The structure and molecular weight of the polymers was characterized by Fourier transform infrared (FTIR) and nuclear magnetic resonance (NMR) techniques. Once polymerized, anion-exchange reactions were carried out to replace the halides with four other counter-anions. Subsequently, the effect of the nature of the anion and the cation on the polymers’ thermal and hygroscopic properties was studied in detail by thermogravimetric analysis (TGA), differential scanning calorimetry (DSC), and FTIR spectroscopy. Depending on the chemical structures of the polymeric cations and the related anions, tailored polymers with a glass transition temperature (*T*_g_) in the range of 30 °C to 131 °C and a thermal stability varying between 170 °C and 385 °C were obtained.

## 1. Introduction

Due to their remarkable properties (e.g., ion conductivity, flame retardancy, thermal stability, and ability to undergo bond exchange reactions), ionic liquids have become attractive building blocks in the design of functional polymers and polymer networks [1,2]. Having a large organic cation and an organic or inorganic counter-anion [3,4], they are characterized by melting points below 100 °C (and in some cases even below room temperature) [5]. The melting temperature decreases with increasing size, higher cation asymmetry, and weaker interactions between the ions [6]. The strength of ionic interactions is governed by shielding or delocalization of the charges through electron withdrawing groups [3,7].

Protic ionic liquids bear positively charged cations, and are easily obtained by neutralization of a Brønsted base with a Brønsted acid [8,9]. In contrast, aprotic ionic liquids are synthesized by an S*n*2 reaction of alkyl halides with the respective base [10]. Aprotic ionic liquids benefit from a higher thermal and chemical stability compared to their protic counterparts. Moreover, the thermal stability decreases with rising hydrophilicity of the anion and the following order is reported for a series of 1-butyl-3-methyl imidazolium-based ionic liquids: PF_6_^−^ > Tf_2_N^−^ > BF_4_^−^ > TfO^−^ > NO_3_^−^ > Br^−^ > Cl^−^ > acetate [11]. Amongst commonly used cations, pyrrolidinium and imidazolium are reported to provide the highest average thermal stability [12].

For the synthesis of polymers comprising covalently bonded ions in their structure, various strategies have been reported. Ionic polymers can be classified by the charge type of the chemically attached ion (i.e., cationic, anionic or ampholytic). In polyelectrolyte complexes, the counterions are also polymeric. The nature of the immobilized and moveable counterions is crucial for the design of ion exchange resins, membranes, and polyelectrolytes [13]. As an example, lithium conductive polyelectrolytes should comprise a polyanion to achieve meaningful ion conductivity [14]. In contrast, ion exchange resins for desalination require both polyanion and polycation beads to exchange both ions of the respective salts to H^+^ and OH^−^.

Ionic polymers can be further categorized by considering charge density (calculating the ion content in relation to the covalent bonds). Ionic polymers with very low ion content are commonly termed ionomers [13]. Along with the content, the position of the ions provides diversification of ionic polymers. In polyelectrolytes, the ionic groups are located at the side chain, whilst in ionenes, they are part of the polymer backbone [15]. Additional sub-categories also exist such as poly(ionic liquids), where the polymer is derived from ionic liquid monomers, and ionogels, where ionic crosslinks are formed in the presence of a so-called gelator [16,17]. In the literature, the terminology is frequently used interchangeably.

Poly(ionic liquids) are typically obtained by polymerization (e.g., radical-mediated chain growth reaction) of functional monomers bearing the ionic liquid as a pendant group [18,19]. Other synthesis routes involve the preparation of block copolymers of ionic liquid-functional monomers with non-ionic monomers [20,21] or grafting of the ionic liquid group to the polymer backbone [22,23,24,25]. In addition, covalently 3D crosslinked ionic polymer networks are prepared by using acid-protonated amines as hardeners [26] or crosslinkable ionic liquids as multi-functional monomers [27,28,29]. Poly(ionic liquids) are hydrophobic [30,31], antibacterial [32], and non-flammable [33], which make them an interesting material class for various application fields including antistatic coatings, solid state electrolytes, gas separation membranes, or ion exchange resins. Poly(ionic liquids) typically exhibit rather low glass transition temperatures (*T*_g_) as they are often obtained from monomers with only one polymerizable group (to ensure high mobility and ion conductivity) [23,34]. Higher *T*_g_ (>100 °C) materials are typically obtained by chemical crosslinking [35]. However, both a high *T*_g_ and a sufficiently high toughness are crucial when it comes to the application of ionic polymers in mobility and energy applications (e.g., as liner material in hydrogen storage tanks).

Herein, a library of ionenes was synthetized by following an S*n*2 step growth reaction of imidazolium-based ionic liquids across organic halides. The influence of the cation as well as the anion structure on the thermal properties of the ionenes was studied comprehensively. It was shown in previous work that material properties (e.g., mechanical performance and solubility) of thermoplastic poly(ionic liquids) can be easily adjusted by anion exchange after polymerization [36,37,38]. Depending on the monomer structure, some anions result in a significant increase in brittleness, which compromises the processability of the poly(ionic liquids) [39,40]. Shaplov et al. prepared a series of polymethyl methacrylate (PMMA) polymers with varying ionic liquid cations in their side chain whilst the Tf_2_N^−^ anion was kept constant. It was found that the aromatic methyl-imidazolium group yields polymers with a low *T*_g_ of 28 °C, while the polymer with a tri-butyl-ammonium exhibited a *T*_g_ of 80 °C. However, both polymers only formed weak, brittle films [41]. In another study, Tome et al. [38]. exchanged the anion of a commercially available poly(allyl dimethylammonium) chloride with different cyanides. The resulting poly(ionic liquids) varied in their *T*_g_ by up to 52 °C.

Kammakakam et al. synthesized mendable aromatic polyamides with imidazolium moieties in their repeating units [42]. Along with a rapid healing performance (30 s at 60 °C), the materials showed remarkably high thermal (*T*_g_ = 130 °C) properties.

In the present study, we show that by chemical design (molecular structure of the polymeric cations and chemical nature of the anions), the *T*_g_ of ionenes are tailored between 30 °C and 131 °C. Selected high-*T*_g_ ionenes with BF_4_^−^ as the anion are also characterized by high thermal stability and a melt flow index that makes them processable by extrusion. This is of particular interest when it comes to an industrial processing of ionenes for gas separation membranes with selective permeation properties or for solid state electrolytes. Herein, the targeted use case for the synthetized ionenes was a potential liner material for type IV hydrogen pressure tanks, as ionic groups are well known to improve gas barrier properties of polymers [43]. The key criteria are a low hydrogen permeability, maintaining structural integrity under pressure cycles during service, and achieving high thermal stability for operation across typical automotive conditions [44,45]. Moreover, the polymer should be characterized by a low water uptake to ensure a short drying process of the vessel after hydrostatic pressure tests carried out in industry. Thus, the most promising candidate (in terms of low moisture uptake and high thermal stability) of the synthetized ionenes is characterized for its mechanical properties and benchmarked with commercial thermoplastics currently used as liner material [46,47].

## 2. Results

### 2.1. Synthesis of Ionenes

For the use of ionenes in applications such as the mobility or energy sectors, a high thermal stability (*T*_g_ and degradation temperature) combined with a low water uptake/solubility in water is crucial. Herein, four different types of ionenes were synthetized differing in their polymer backbone (e.g., aliphatic versus aromatic building blocks and introduction of amide functions). The base structures of the ionenes are displayed in Figure 1.

1,1′(1,6-hexane)bisimidazole (**M1**) and 1,4-benzenedicarboxamide *N*^1^,*N*^4^-bis[3-(1H-imidazole-1-yl) propyl] (**M2**) were synthetized as aliphatic and aromatic bisimidazole monomers. **M1** was obtained in high yields (>90%) while the yield of **M2** was only 73%. ^1^H-NMR and ^13^C-NMR spectra were in good agreement with the chemical structures and are shown in Appendix A.

Ionenes are typically obtained by the Menshutkin reaction following an S*n*2 step growth reaction between ditertiary amines and dihalides (X-R′-X) [48,49,50,51]. The polymerization yields ionenes with bromide or chloride anions, which are well soluble in water. Subsequent anion exchange reactions with different salts via metathesis are a convenient approach to tune solubility, thermal, and mechanical properties of the ionenes in a post-modification step [52,53,54].

In previous work, Carlisle et al. reported on the synthesis of aliphatic main-chain imidazolium polymers [53]. Herein, this synthesis protocol was employed to prepare aliphatic ionenes with varying counter-anions. In particular, **I1** was obtained by reacting 1,6-dibromohexane across **M1**. The FTIR and ^1^H-NMR spectra corresponded to the structure of the polymerized **I1** and are provided in Figure 2 and Appendix A, respectively. The signal related to the C-H group of the imidazole ring is shifted from 7.60 to 8.60 ppm due to the quaternization of the imidazole giving rise to the progress of the reaction [55]. The small remaining signal at 7.60 ppm is related to the imidazole-based end-groups of **I1**, and was exploited to calculate the molecular mass of the obtained polymer (*M*_w_ ~ 14,500 g/mol).

To study the influence of the anion on thermal properties and water uptake, the bromide of **I1** was exchanged with a series of sodium and lithium salts including Na-*p*-toluene sulfonate (Na-pTSA), NaC(CN)_2_, NaBF_4_, and LiTf_2_N. The FTIR spectra of the obtained polymers are provided in Figure 2 and compared to the FTIR data of the salts used for the anion exchange. After the anion exchange, the products were dried at 60 °C under vacuum overnight. While **I1** and **I1-BF_4_** appeared as a white powder, **I1-TF_2_N**, **I1-pTSA**, and **I1-C(CN)_2_** were sticky. In the FTIR spectra of **I1**, **I1-C(CN)_2_**, and **I1-pTSA**, a broad -OH band is observed between 3550 and 3150 cm^−1^, which indicates the presence of residual H_2_O. From TGA experiments (Figure 3b) it can be obtained that the moisture uptake ranges from 3 wt% (**I1-pTSA**) to 5 (**I1**) and 6 wt% (**I1-C(CN)_2_**). However, extending the drying time for an additional 12 h did not lead to a significant change—neither in the appearance of the polymers nor in their FTIR spectra. In contrast, the moisture content of **I1-TF_2_N** is well below 1 wt% according to the TGA data.

Independent of the type of anions, absorption bands between 3200–2800 cm^−1^ were detected, which are related to stretching vibrations of N-H and C-H groups [56,57,58]. Additionally, the N-H stretching bands of the imidazole ring arose at 2940 cm^−1^ and 2861 cm^−1^. The successful anion exchange with C(CN)_2_^−^ was evidenced by the appearance of new peaks at 2286 cm^−1^, 2227 cm^−1^, and 2149 cm^−1^ related to C≡N vibration bands [56,57]. For **I1-pTSA**, the characteristic *p*-toluene sulfonate peaks were detected at 1121 cm^−1^, 1011 cm^−1^, and 678 cm^−1^, while for **I1-Tf_2_N**, new peaks appeared at 1346 cm^−1^ and 1050 cm^−1^. With respect to **I1-BF_4_**, the change in the FTIR spectrum was less pronounced but an additional characteristic peak was observed at 1010 cm^−1^, which can be assigned to the stretching vibrations of -BF_4_ bonds (as also observed for the corresponding spectrum of NaBF_4_) [59].

**I2** as an aromatic ionene with amide groups was synthetized by following a condensation reaction between 1,6-dibromohexane and **M2** as a diamide-containing aromatic bisimidazole [50,54]. The ^1^H-NMR spectrum is displayed in Appendix A and corresponds well with the structure of the water-soluble polymerized product. The shift of the methylene protons of the imidazole ring is assigned to the quaternization of the imidazole groups, whereas the C-H of the free imidazole end-groups at 6.92 ppm [55,60] was used to calculate the molecular weight, which was significantly lower (Mw ~ 6000 g/mol) compared to **I1**. The C-H signals of the terephthaloyl group of the imidazole-based monomer remained unchanged and were observed at 4.02 ppm, 3.14 ppm, 1.93 ppm, 1.46 ppm, and 0.96 ppm. It should be noted that the anion exchange of **I2** was only performed with NaBF_4_, due to the superior performance of BF_4_^−^-based ionenes, which will be discussed in the following section. The exchange of the anion was confirmed by FTIR measurements. The FTIR spectra of **I2** and **I2-BF_4_** in comparison with NaBF_4_ are displayed in Figure 4. The signals in the range of 3600–2800 cm^−1^ were assigned to overlapping stretching vibrations of -OH, C-H, and N-H bonds [56,57,58]. The characteristic stretching vibrations of the -BF_4_ bonds are observed at 1010 cm^−1^. In addition, the bands at 1655 cm^−1^, 1538 cm^−1^, 1490 cm^−1^, and 1449–1441 cm^−1^ were assigned to the stretching vibration of C=C and C=N bonds of the imidazole rings.

For the FTIR spectrum of **I2-BF_4_**, no significant -OH band can be observed, which also correlates with the TGA data showing no significant weight loss between 100 and 200 °C. In contrast, a broad -OH band between 3550 and 3150 cm^−1^ related to residual water is evident in the FTIR spectrum of **I2**. According to the TGA data, **I2** exhibits a moisture uptake of about 3 wt% prior to the anion exchange.

To further increase the number of aromatic groups in the ionene, **I3** was synthetized by a condensation reaction between **M2** and 1,4-bis(chloromethyl)benzene. The ^1^H-NMR spectrum is shown in Appendix A. The methylene protons of the imidazole ring are again shifted to 7.60 ppm, confirming the quaternization of the imidazole. A new peak at 5.40 ppm was detected, which is related to the C-N link formed by the *Sn2* reaction. This is in good agreement with the work of Bara and co-workers [61]. The C-H signals at 7.03 ppm and 6.92 ppm were assigned to the H-atoms of the free imidazole rings at the terminal end of the polymer chain and used to estimate the molecular weight (*M*_w_ ~ 30,000 g/mol). Anion exchange was performed with NaBF_4_ yielding **I3-BF_4_**, whose FTIR spectrum is provided in Figure 5 in comparison with **I3** and NaBF_4_ as reference. In **I3-BF_4_**, the typical signal related to the -BF_4_ bonds could be detected at 1013 cm^−1^, whilst the distinctive -OH band (observed for **I3** between 3550 and 3150 cm^−1^) is missing. From the corresponding TGA curves (Figure 3a) it can be obtained that the moisture uptake decreases from 6 to 1 wt% by replacing the chloride groups with BF_4_^−^.

Finally, **I4** was prepared by following a condensation reaction between **M2** and 1-chloro-4-(4-chlorophenyl)benzene. The ^1^H-NMR spectrum is provided in Appendix A. While the C-N link formed by the reaction is confirmed by the signal arising at 5.30 ppm, the expected shift of the imidazole signals due to quaternization could not be detected. This might be explained by the replacement of hydrogen atoms with deuterons of the solvent and might have happened during the storage of a few days prior to the NMR measurements [62]. For the estimation of the molecular weight via end-group analysis, the overlapping C-H peaks in the range of 7.8 ppm to 7.07 ppm were used (*M*_w_ ~ 27,500 g/mol). **I4-BF_4_** was then obtained again by anion exchange with NaBF_4_. Figure 6 shows the FTIR spectra of **I4** and **I4-BF_4_** in comparison with NaBF_4_. The absorption band assigned to the -BF_4_ bonds was clearly observable at 1010 cm^−1^. In addition, in the FTIR spectrum of **I4**, the broad -OH band related to residual water appeared between 3550 and 3150 cm^−1^. However, the amount of water uptake could not be determined from the respective TGA curve quantitatively, as the sample also contained solvent residues (2-methoxy-2-methylpropane), which started to evaporate at 55 °C.

### 2.2. Thermal Properties of Ionenes

The thermal stability of the ionenes under investigation was determined by TGA analysis. Previous studies revealed that the majority of the degradation processes of ionic liquids involve *SN2* and *SN1* pathways [11]. In imidazole-based ionic liquids, cleavage of the C-N bond proceeds by the anion attacking the cation. In addition, elimination or rearrangement reactions are reported for ionic liquids bearing non-coordinative anions [63]. Comparing the temperature at 5% of mass loss (*T*_5%_) and the onset temperature for the major weight loss (*T*_onset_) of the different ionenes, the results clearly show that the thermal stability is mainly governed by the anion whilst the cation plays a minor role (Table 1).

For example, **I1-BF_4_**, **I2-BF_4_**, **I3-BF_4_**, and **I4-BF_4_** comprised *T*_onset_ values of 330 °C, 332 °C, 345 °C, and 342 °C, respectively (Figure 3a and 3b). In contrast, the *T*_onset_ value differed by more than 250 °C when comparing **I1** (*T*_onset_ = 270 °C) with **I1-Tf_2_N** (*T*_onset_ = 398 °C) (Figure 3b). It is reported in the literature that the coordinating nature, nucleophilicity, and hydrophilicity of the anion significantly affect the thermal stability of ionic liquids [11].

This trend is also found in the ionenes under investigation, showing a lower thermal stability for ionenes that are more prone to water uptake. In the FTIR spectra of **I1**, **I1-C(CN)_2_**, and **I1-pTSA**, the characteristic -OH bands of water could be detected, which is also reflected in the gradual weight loss between 100 and 200 °C in the respective TGA curves. These hydrophilic ionenes have a lower thermal stability than **I1-BF_4_** and **I1-TF_2_N**, which comprise fewer nucleophilic anions.

A shift towards higher thermal stability can be also observed in **I2**, **I3**, and **I4** when exchanging the bromide or chloride anion with BF_4_^−^. The stabilizing effect is particularly pronounced for **I4**, for which *T*_onset_ increased from 185 to 342 °C. Here, the exchange of chloride with BF_4_^−^ even changes the degradation mechanism from a two-stage to a one-step process.

In the second step, the glass transition temperature (*T*_g_^onset^) of the ionenes was studied to evaluate possible application fields. Figure 7a displays the DSC curves of **I1** with varying anions and it can be clearly seen that the anion does not only affect thermal stability but also *T*_g_ values of the ionenes. It should be noted that the *T*_g_ values were derived from the second heating run as residual water is evaporated in the first heating run. This also leads to a slight shift to higher *T*_g_ values as residual water can act as lubricant [64,65,66,67]. Due to the aliphatic nature of **I1**, the *T*_g_ values are well below 100 °C, which makes them unsuitable for any structural application in automotive industry.

It is obvious that the *T*_g_ onset values are not correlated with the TGA data and the least stable **I1** gave the highest *T*_g_ (55 °C). The results revealed that the *T*_g_^onset^ decreased with increasing size of the anion and amounted to −18 °C, 45 °C, 16 °C, and −31 °C for **I1-pTSA**, **I1-BF_4_**, **I1-C(CN)_2_**, and **I1-TF_2_N**, respectively.

By increasing the number of aromatic groups in the polymer backbone, the *T*_g_ value could be significantly increased (Figure 7b). By replacing the aliphatic imidazole **M1** in **I1** with **M2** having one aromatic ring, **I2** was obtained, which had a *T*_g_ of 120 °C. However, exchanging the bromide with BF_4_^−^ led again to a drop in the *T*_g_ (<100 °C). This trend was also observed for **I3** albeit at a different level. In **I3**, both monomers comprise aromatic groups, which gave an ionene with a *T*_g_ of 165 °C. While the exchange with BF_4_^−^ decreased the *T*_g_, it still amounted to 130 °C. For the synthesis of **I4**, 1,4-bis(chloromethyl)benzene was replaced with 1-chloro-4-(4-chlorophenyl)benzene as co-monomer to further increase the number of aromatic groups. However, the *T*_g_ could not be increased in comparison to **I3** and it reached 125 °C. Surprisingly, the exchange of the chloride with BF_4_^−^ shifted the Tg to slightly higher values (131 °C). A possible mechanism for this behavior is not clear and needs further investigations.

### 2.3. Possible Application as High-Pressure Hydrogen Gas Storage Liner

Ionenes are of great importance for membrane materials due to their enhanced permeability and selectivity towards CO_2_. Since permselectivity is inversely proportional to permeability, those ionic polymers can exhibit significant gas barrier properties to H_2_ in CO_2_/H_2_ separation processes. This behavior presents a potential advantage for use as a liner material in a 70 MPa type IV hydrogen pressure. Such vessels consist of a thermoplastic liner (e.g., polyethylene, PE, or polyamide, PA) that acts as a gas barrier against hydrogen with a composite overwrap that maintains the structural integrity of the vessel under high-pressure conditions. Performance criteria for polymer liner materials focus on low hydrogen permeability, maintaining structural integrity under pressure cycles during service, and achieving high thermal stability for operation across typical automotive conditions in the range of −40 °C to +80 °C. Furthermore, a low affinity to water is beneficial for reducing drying process time of the vessel after the typical hydrostatic pressure test. In order to make initial statements about the suitability of ionene as a lining material, we herein focus on **I4** with the tetra-fluoroborate anion. As there is no universal standard for the required material properties of thermoplastic liner materials for type IV hydrogen storage tanks at low temperatures down to −40 °C, test results from Vicat and tensile tests at room temperature were determined and analyzed against values for PE and PA from the literature and unpublished industrial standards.

Up to 40 g of I_4_-BF_4_- were prepared according to the procedure described in Section 2.1 and formed into defect-free and uniform plates by hot pressing. Steel frames were used to achieve the desired thickness of the plaques, which was 1 and 3 mm. The polymer was ground to uniform particles and then dried under vacuum. Subsequently, the material was placed with an excess of approximately 10% and covered by two Teflon sheets on the preheated (260 °C) lower aluminum plate of the hot press machine (P 200 PV, Dr. Collin, Ebersberg, DE, Germany). After closing the hot press, a pressure of 10 MPa was applied on the material for approximately 1 min. Finally, the pressure was released, and the sandwiched plates were removed and cooled down to room temperature.

Results from Vicat testing shows that the determined softening temperature (140 °C) exceeds the targeted value of >100 °C. This threshold was defined based on literature data for high-density polyethylene (HDPE), with typical Vicat softening temperatures around 120–130 °C, and polyamide (PA6.6), which generally exhibits higher values in the range of 200–220 °C [68]. HDPE and PA6.6 are thermoplastics commonly used as liner materials for composite-based storage tanks [69,70].

The average tensile modulus and strength values are 1357 ± 135 MPa and 20 ± 2 MPa, respectively. The strain at break values at room temperature, however, are not higher than 1.6 ± 0.3%. For comparison, HDPE typically exhibits a tensile modulus of 800–1500 MPa, tensile strength of 20–30 MPa, and strain at break values well above 100%, while PA6.6 generally shows a tensile modulus of 2800–3200 MPa, tensile strength of 75–85 MPa, and strain at break between 20% and 50% [68].

Since an appropriate elasticity is crucial to prevent damage caused by pressure changes in the tank structure, the material has to be further improved to be applicable as a liner material in type IV hydrogen pressure vessels. One way to reduce the brittleness of **I4** is to increase its *M*_w_. It is well reported in the literature that a higher *M*_w_ promotes ductility in thermoplastic materials by contributing to chain entanglements and energy dissipation [71,72].

## 3. Materials and Methods

### 3.1. Materials and Chemicals

Gases were purchased from Linde GmbH (Pullach, Germany) with a purity of at least 99.99%. Sodium hydroxide (≥99%), tetrahydrofuran (THF) (≥99.5%), potassium carbonate_,_ acetonitrile (ACN), (≥99.8%), *N*-methyl-2-pyrrolidone (NMP) (99.8%), dichloromethane (99.8%), and 2-methoxy-2-methylpropane (MTBE) were purchased from Carl Roth GmbH & Co. KG (Karlsruhe, Germany). 1H-imidazole (≥99.5%), 1,6-dibromhexane (DBrH) (≥96%), terephthaloyl chloride (TC) (≥99%), 1-(3-aminopropyl)imidazole (API) (≥97%), 1-chloro-4-(4-chlorophenyl)benzene (dichlorobiphenyl), 1,4-bis(chloromethyl)benzene (*p*-DCX) (98%), natriumtoluene-4-sulfonate (*p*-toluene sulfonate) (95%), basic activated aluminum-oxide, and magnesium sulfate were obtained from Sigma Aldrich Corp. (St. Louis, MO, USA). Sodium dicyanamide (95%), sodium tetrafluoroborate, and lithium-bis(trifluormethylsulfonyl)imide salt (99%) (99.8%) were supplied by ABCR (Karlsruhe, Germany). ACN and NMP were dried with 3A and 4A molecular sieves, respectively, before further usage. All other mentioned chemicals were used without any further purification.

### 3.2. Synthesis of Monomers

#### 3.2.1. Synthesis of 1,1′(1,6-Hexane)bisimidazole (**M1**)

**M1** was prepared according to the synthesis protocol described by Bara et al. [61]. As a first step, sodium imidazolate (NaIm) was prepared under dry conditions by placing 15.01 g (220 mmol) 1H-imidazole and 9.26 g (231.4 mmol) of finely grounded NaOH in a round-bottom flask. The flask was sealed with a rubber plug and the mixture was continuously stirred for 5 h at 110 °C until it turned into a yellowish liquid. Subsequently, the reaction was stopped and the received product was dried in the vacuum oven overnight and characterized via FTIR and ^1^H-NMR. Yield: 44.08 g (90.3%). ^1^H-NMR (δ, 300 MHz, DMSO-*d*_6_): 6.99 (s, 1H, C^a^), 6.59 (d, 2H, C^b,c^) ppm.

In the second step, 5.14 g NaIm, (169 mmol), 0.45 mol equivalent of 1,6-dibromhexane, and 40 mL of THF were placed into a round-bottom flask and stirred at room temperature for 3 h. Once a stable dispersion was obtained, the temperature was raised to 67 °C and the mixture was stirred under reflux overnight. After the reaction was completed, the heat was switched off, and the mixture was cooled while stirring for at least one hour. Subsequently, the mixture was filtered through a filter paper and washed with THF. Then the combined liquid phases were reduced via rotary evaporation and 100 mL of CH_2_Cl_2_ was added to the product. Quantities of 4.00 g of MgSO_4_ and 4.00 g of activated carbon were added to the mixture, which then was stirred for 5 min. Afterwards the product was filtered through a plug of basic activated Al_2_O_3_ to receive a yellowish liquid. The solids were washed with 100 mL of CH_2_Cl_2_ and then the liquid phase was reduced via rotary evaporation. The product (**M1**) was dried at 40 °C under vacuum overnight before the flask was flushed with nitrogen, sealed, and then stored in the fridge. Yield: 15.23 g (91.6%). ^1^H-NMR (δ, 300 MHz, DMSO-*d*_6_): 7.60 (s, 2H), 7.16 (s, 2H), 6.88 (s, 2H), 3.93 (t, 4H), 1.78 (p, 4H), 1.22 (p, 4H) ppm.

#### 3.2.2. Synthesis of 1,4-Benzenedicarboxamide *N*^1^,*N*^4^-Bis[3-(1H-imidazole-1-yl) Propyl] (**M2**)

**M2** was synthetized using a procedure similar to that published by Bara et al. [54]. For this, 2.922 g API (23.3 mmol), 4.40 g K_2_CO_3_ (31.93 mmol), and 5 mL ACN were added to a three-necked round-bottom flask. The flask was mounted on a magnetic stirrer equipped with an oil bath, a reflux condenser, and a bubbler. The mixture was stirred for 30 min at room temperature and purged with nitrogen gas. A quantity of 2.154 g TC (10.51 mmol) was then dissolved in 15 mL ACN and slowly added to the reaction vessel via a syringe, before the nitrogen flow was turned off and the reaction was heated to reflux and stirred for 24 h. After the mixture was cooled down, 100 mL of de-ionized water was added and the mixture was then stirred at 40 °C for 2 h. The reaction product was then filtered through a glass frit and washed with 50 mL de-ionized water (H_2_O). The solids were collected and dried at 100 °C for 24 h under vacuum. Yield: 2.935 g (73%). ^1^H-NMR (δ, 300 MHz, DMSO-*d*_6_): 8.66 (t, 2H), 7.93 (s, 4H), 7.68 (s, 2H), 7.23 (s, 2H), 6.91 (s, 2H), 4.04 (t, 4H), 3.25 (q, 4H), 1.99 (p, 4H) ppm.

### 3.3. Synthesis of Ionenes

#### 3.3.1. Synthesis of Ionene 1 (**I1**)

The copolymerization of **M1** and 1,6-dibromohexane yielding **I1** was slightly adapted from the procedure reported by Carlisle et al. [53]. Into a round-bottom flask equipped with a magnetic stir-bar, **M1** was diligently placed and dissolved with 5–10 mL ACN. Into a 50 mL beaker, 1 mol equivalent of a 1,6-dibromohexane was placed, dissolved in 5–10 mL of solvent, and added to the imidazolium monomer **M1** while stirring. Then the mixture was heated to 85 °C for 96 h under reflux. During this process the polymer precipitated on the inner walls of the flask. Following this, the reaction was cooled to room temperature, then extracted with 2 mL of solvent and precipitated in 40–60 mL of MTBE. The product was washed two times with MTBE and residues of the solvent were removed via rotary evaporation. The obtained polymer (Mw ~ 14,500 g/mol) was dried under vacuum at 55 °C for 24 h. Yield: 3.12 (98.1%). ^1^H-NMR (300 MHz, DMSO-*d*_6_): 8.64 (s), 7.34 (s), 4.01 (t), 1.72 (s), 1.20 (s).

#### 3.3.2. Synthesis of Ionene 2 (**I2**)

**I2** was obtained by copolymerization between **M2** and 1,6-dibromohexane following the synthesis protocol of Bara et al. [54]. Firstly, the **M2** and 1,6-dibromohexane were added in an equimolar amount to a 50 mL round-bottom flask. A quantity of 10–15 mL of NMP was then added, and the flask was sealed with a rubber cap. Once **M2** was completely dissolved, an injection needle was inserted through the rubber cap to equalize the pressure. When the reaction was completed, the heat was switched off and the mixture was cooled to room temperature for 1 h. The inner walls of the flask and the stir-bar were coated by the polymer, as it had precipitated from the solution during the proceeding polymerization. NMP was decanted, the polymer washed 2 times with 30–40 mL of MTBE and the solvent was removed via rotary evaporation. Finally, the polymer (Mw ~ 6000 g/mol) was dried in the vacuum oven between 100 and 155 °C. Yield: 0.734 g (93.5%). ^1^H-NMR (δ, 300 MHz, DMSO-*d*_6_): 8.59 (s), 7.47 (d), 7.24 (s), 7.11 (s), 4.02 (t), 3.14 (t), 1.93 (t), 1.46 (t), 0.96 (t) ppm.

#### 3.3.3. Synthesis of Ionene 3 (**I3**)

**I3** (Mw ~ 35,500 g/mol) was obtained by following the procedure described in Section 3.2.2, while 1,6-dibromohexane was replaced with 1,4-bis(chloromethyl)benzene as co-monomer. Yield: 0.98 g (89.8%). ^1^H-NMR (δ, 300 MHz, DMSO-*d*_6_): 8.90 (s), 7.77 (s), 7.44 (m), 7.03 (s), 5.40 (s), 4.33 (t), 3.49 (t), 2.83 (s), 2.33 (m) ppm.

#### 3.3.4. Synthesis of Ionene 4 (**I4**)

**I4** (Mw ~ 27,500 g/mol) was obtained by following the procedure described in Section 3.2.2 while 1,6-dibromohexane was replaced with 1-chloro-4-(4-chlorophenyl)benzene as co-monomer. Yield: 1.80 g (81.1%). ^1^H-NMR (300 MHz, DMSO-*d*_6_): 7.90–6.9 (m, 18H), 5.31 (s, 2H), 4.14 (s, 2H), 3.20 (s, 4H), 2.31 (s, 4H) ppm.

#### 3.3.5. Anion Exchange

The anion exchange of the synthetized ionenes **I1**–**I4** was adapted from already published synthesis protocols [53,54,73]. In particular, NaBF_4_, sodium-*p*-toluene sulfonate, NaC(CN)_2_, and LiTf_2_N were utilized as anion exchange salts. The ionene was dissolved while stirring at a ratio of 1 g polymer per 10–15 mL of de-ionized H_2_O in a 25 mL round-bottom flask at room temperature. A quantity of 2.1 mol equivalent of the salt solution (1 g salt per 5–10 mL H_2_O) was added dropwise to the dissolved polymer at a rate of 3–6 drops per minute, whereby a white or brownish precipitate was observed immediately. After the whole salt solution was added, the mixture was stirred for an additional 20 min before filtering and washing 2 times with 5–10 mL of H_2_O. The product was filtered and dried under vacuum at 70–155 °C for 24 to 48 h. Finally, the flask with the product was flushed with nitrogen, sealed, and the stored in the fridge.

### 3.4. Characterization Methods

^1^H and ^13^C nuclear magnetic resonance spectra (NMR) were recorded on an Avance III 300 MHz spectrometer (Bruker, Billerica, MA, USA) with dimethylsulfoxide-*d*_6_ (DMSO-*d*_6_) as solvent and trimethylsilane as internal standard. The NMR spectra were processed with the software MestreNova Version: 10.0.2 (MestreLab Research, Santiago de Compostela, Spain). Additionally, an end-group calculation was performed to estimate the approximate molecular weight (M*_w_*) of the polymers [74,75,76].

FTIR spectra were recorded on a Spectrum Two (Perkin Elmer, D, Waltham, MA, USA) in attenuated total reflectance (ATR) mode with the software Spectrum Version 10.6.2 (Perkin Elmer, D). The applied spectral range was from 4000 cm^−1^ to 650 cm^−1^ with a total number of 8 scans and a resolution of 4 cm^−1^.

Differential scanning calorimetry (DSC) measurements were performed on a Perkin Elmer DSC 6000 (Perkin Elmer, Waltham, MA, USA) and a Mettler-Toledo DSC1 (Mettler-Toledo, Columbus, OH, USA). The measurements were carried out with a heating rate of 20 K/min under nitrogen atmosphere. For the DSC characterization, 10–14 mg of the samples was weighed into 40 µL standard aluminum pans. All samples were first heated from −30 °C to 200 °C, then cooled to the start temperature of the first run and again heated to 200 °C.

Thermogravimetric analysis (TGA) measurements were performed on a Mettler-Toledo TGA/DSC (Mettler-Toledo, Columbus, OH, USA) by heating the samples from 25 °C to 600 °C (with a heating rate of 10 K/min) under nitrogen and from 600 °C to 900 °C under oxygen. The samples were weighed in a range of 10–18 mg into 70 µL standard ceramic crucibles. The evaluation was performed with the STAR^e^ Evaluation software version 12.1 (Mettler-Toledo, Columbus, OH, USA).

Water jet cutting was employed to cut the plaques into standardized test specimens for tensile and VICAT testing. Prior to testing, all samples were dried in a heating oven at 150 °C for at least 8 h. Quasi-static tensile tests were performed at room temperature in accordance with EN ISO 527-1 [77] on a universal tensile/compression test machine (Z001, Zwick, Ulm, Germany) equipped with a 1 kN load cell. Specimens were loaded via pneumatic grips at a test speed of 1 mm/min. At least 5 specimens (EN ISO 527-2 type 5B) [78] were used for the quasi-static tensile tests to determine tensile modulus, tensile strength, and strain at brake (Q-tec GmbH, Zeilarn, Germany). Measurements for determining the Vicat softening temperature were performed in accordance with ISO 306 [79] using the Vicat softening point tester HDT/VICAT 6510 (Instron CEAST Division, Pianezza, Italy) under the load of 50 N and a heating rate of 120 °C/h from room temperature until the sample deformed by 1 mm. Two specimens with a thickness of 3 mm and 9.5 mm square geometry were tested.

## 4. Conclusions

A series of ionenes was synthetized with varying types of cations and anions, and their influence on thermal stability, water uptake, and *T*_g_ was studied. It was found that the type of anion governs hydrophilicity (water solubility) and degradation temperature of ionenes with higher polar ionenes having a lower thermal stability. In contrast, the chemical nature of the cations within the polymer backbone has a lesser effect on both water solubility and degradation temperature. However, it significantly governs the *T*_g_ of the ionenes. By introducing aromatic groups in the polymer backbone, the *T*_g_ can be shifted up to 158 °C. However, bulky anions giving a high thermal stability compromise the *T*_g_ and lead to a decrease. **I4** is here an exception, but a possible explanation for this behavior needs additional studies. For using ionenes in technical applications such as barrier materials in automotive applications, a balance between high *T*_g_ (at least 105 °C), high temperature stability, and low water uptake is required. **I4-BF_4_** fulfills these criteria. Hence its suitability as liner material in type IV hydrogen was further analyzed. Hot press technology has been successfully used to produce test specimens for tensile testing and heat resistance (Vicat). The material shows sufficient values regarding the Vicat softening temperature, but tensile tests at room temperature have demonstrated that **I4-BF_4_** is too brittle for an application as liner material in type IV hydrogen vessels. However, further optimization of the structure of **I4-BF4** or blending it with established polymer liner materials such as PA6 or PA11 could open up possibilities for using this class of materials in future applications of energy storage.

## Figures and Tables

**Figure 1 molecules-30-03961-f001:**
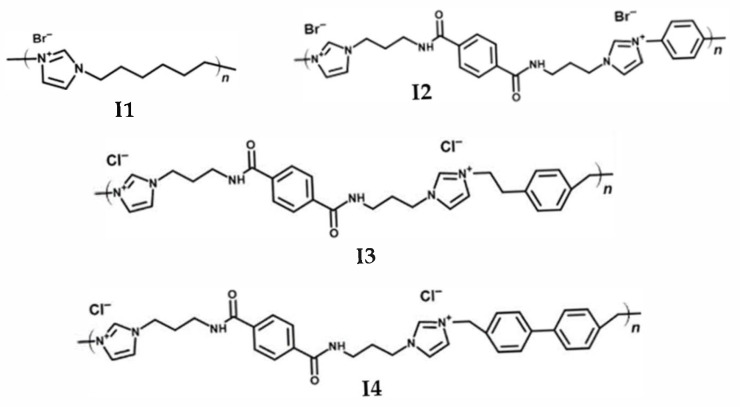
Molecular structure of synthetized ionenes.

**Figure 2 molecules-30-03961-f002:**
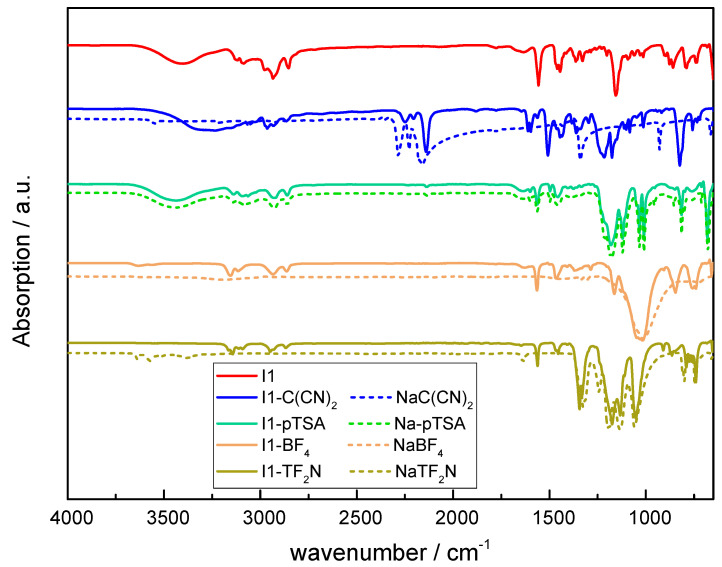
ATR-FTIR spectra of **I1** prior to and after exchange with selected anions (solid lines) and related salts used for anion exchange as reference (dotted lines).

**Figure 3 molecules-30-03961-f003:**
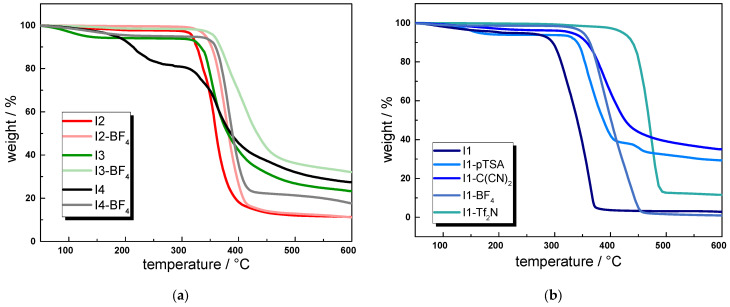
TGA curves of ionenes under investigation showing the weight loss versus temperature: (**a**) influence of the structure of the cation on thermal stability of ionenes; (**b**) influence of the structure of the anion on the thermal stability of ionenes.

**Figure 4 molecules-30-03961-f004:**
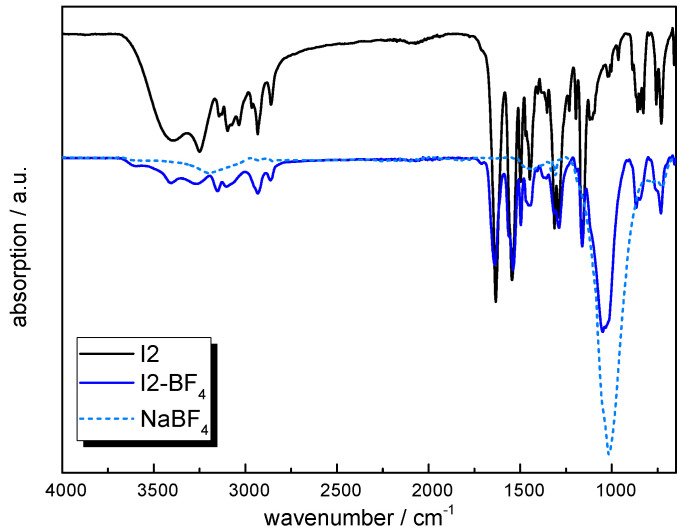
ATR-FTIR spectra of **I2** prior to and after exchange with NaBF_4_ (solid lines) and spectrum of NaBF_4_ as reference (dotted lines).

**Figure 5 molecules-30-03961-f005:**
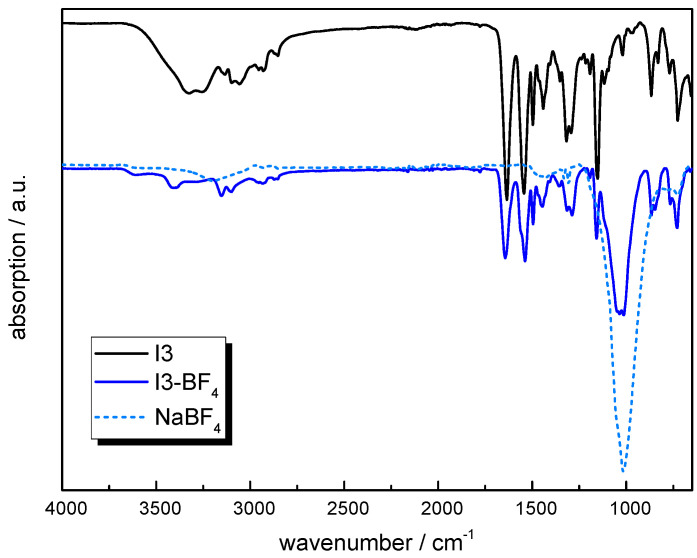
ATR-FTIR spectra of **I3** prior to and after exchange with NaBF_4_ (solid lines) and spectrum of NaBF_4_ as reference (dotted lines).

**Figure 6 molecules-30-03961-f006:**
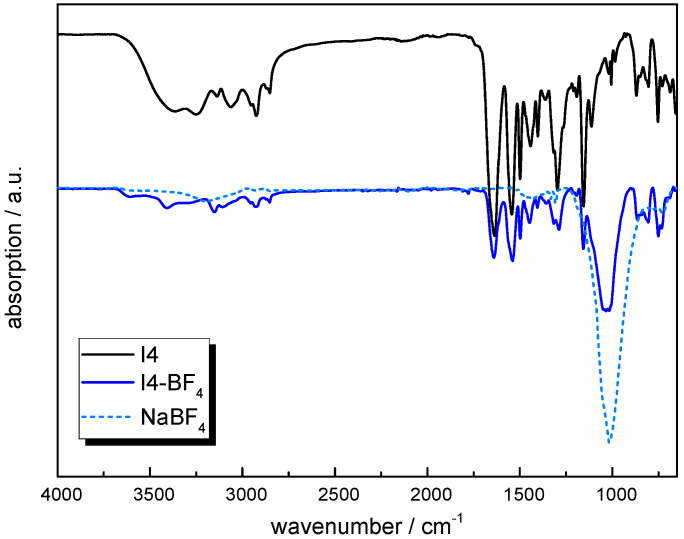
ATR-FTIR spectra of **I4** prior to and after exchange with NaBF_4_ (solid lines) and spectrum of NaBF_4_ as reference (dotted lines).

**Figure 7 molecules-30-03961-f007:**
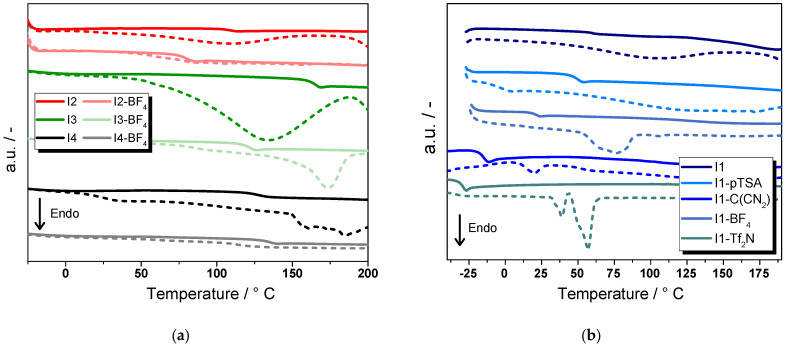
DSC curves of ionenes under investigation showing the heat flow versus temperature: (**a**) influence of the structure of the cation on thermal stability of ionenes; (**b**) influence of the structure of the anion on the thermal stability of ionenes. Dashed lines denote the first heating run and solid lines the second heating run from which the *T*_g_ was determined.

**Table 1 molecules-30-03961-t001:** Summary of the thermal properties (*T*_5%_, *T*_onset_, and *T*_g_^onset^) of the ionenes under investigation.

Ionene	*T*_5%_ (°C)	*T*_onset_ (°C)	*T*_g_^onset^ (°C)
**I1**	283	270	55
**I1-pTSA**	163	323	45
**I1-C(NCN)_2_**	351	327	−18
**I1-BF_4−_**	330	330	16
**I1-TF_2_N**	414	398	−31
**I2**	318	312	105
**I2-BF_4−_**	342	332	75
**I3**	110	320	158
**I3-BF_4−_**	356	345	116
**I4**	183	185	123
**I4-BF_4−_**	269	342	131

## Data Availability

Data are available on request.

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
