# Peer review of "Synthesis and Characterization of Imidazolium-Based Ionenes"

_molecules, 2025, doi:10.3390/molecules30193961_

Round 1

Reviewer 1 Report

Comments and Suggestions for Authors

A series of ionenes was synthetized with varying types of cations and anions, and their influence on thermal stability, water uptake and Tg was studied. It was found that the type of anion governs hydrophilicity (water solubility) and degradation temperature  of ionenes with higher polar ionenes having a lower thermal stability. The paper was interesting and suitable for publication in Molecules with the following changes:

  • The pictures relative to the FTIR analysis have to be included in the paper and not in the supporting information because the discussion on the peak assignments is difficult to follow for the reader.
  • Although an example of application of the proposed research it should be better to detail better in the introduction the fields in which this research can be applied to n give improvement of technical relevance.

Author Response

A series of ionenes was synthetized with varying types of cations and anions, and their influence on thermal stability, water uptake and Tg was studied. It was found that the type of anion governs hydrophilicity (water solubility) and degradation temperature  of ionenes with higher polar ionenes having a lower thermal stability. The paper was interesting and suitable for publication in Molecules with the following changes:

Comment 1: The pictures relative to the FTIR analysis have to be included in the paper and not in the supporting information because the discussion on the peak assignments is difficult to follow for the reader.

Response 1: As suggested, the FTIR spectra were moved from the supporting information to the main document (Figure 2-5 in revised manuscript).

Comment 2: Although an example of application of the proposed research it should be better to detail better in the introduction the fields in which this research can be applied to give improvement of technical relevance.

Response 2: The targted use case and aditional examples of possible fields of applications were included in the introduction part (page 3, second paragraph).

Reviewer 2 Report

Comments and Suggestions for Authors

This manuscript presents a well-structured and comprehensive study on the synthesis, characterization, and thermal analysis of a series of imidazolium-based ionenes with varying cationic and anionic structures. The work is scientifically sound, methodologically rigorous, and clearly presented. The authors successfully demonstrate the tunability of thermal properties through structural modifications and anion exchange, and they provide a preliminary evaluation of the material’s potential application as a liner for hydrogen storage tanks. I recommend major revisions prior to publication, as outlined below.

  1. The discussion of hydrophilicity is supported by FTIR and TGA, but direct water uptake measurements would provide quantitative support.
  2. Discuss how molecular weight affects properties like brittleness.
  3. While FTIR confirms anion exchange, quantitative analysis (e.g., elemental analysis or ICP-OES) would provide more definitive evidence of complete anion replacement, especially for anions like BF₄⁻ where spectral changes are subtle.
  4. While the preliminary evaluation of I4-BF₄ as a liner material is valuable, a more direct comparison with commercial materials would strengthen the application argument.

Minor Points

  1. Define all abbreviations upon first use, e.g., pTSA.
  2. In the supplementary materials, the 13C NMR spectra should be provided.
  3. page 6, line 220, boride should read bromide.
  4. References 50 and 51 miss specific chapter authors or page numbers.

Author Response

This manuscript presents a well-structured and comprehensive study on the synthesis, characterization, and thermal analysis of a series of imidazolium-based ionenes with varying cationic and anionic structures. The work is scientifically sound, methodologically rigorous, and clearly presented. The authors successfully demonstrate the tunability of thermal properties through structural modifications and anion exchange, and they provide a preliminary evaluation of the material’s potential application as a liner for hydrogen storage tanks. I recommend major revisions prior to publication, as outlined below.

Comment 1: The discussion of hydrophilicity is supported by FTIR and TGA, but direct water uptake measurements would provide quantitative support.

Response 1: Some of the polymers are fully soluble in water, which makes a comparison for the moisture uptake difficult. Consequently, we calculated the moisture uptake from TGA experiments to provide quantitative data and included the results in the discussion part (see chapter 2.1 in revised manuscript).

Comment 2: Discuss how molecular weight affects properties like brittleness.

Response 2: An additional paragraph was included in the discussion part highlighting the influence of molecular weight on brittleness (page 10, paragraph 2).

Comment 3: While FTIR confirms anion exchange, quantitative analysis (e.g., elemental analysis or ICP-OES) would provide more definitive evidence of complete anion replacement, especially for anions like BF₄⁻ where spectral changes are subtle.

Response 3: We agree that further analytical tools would give a quantitative insight into the number of groups replaced. However, we observed distinctive changes in polarity, thermal stability and thermal properties along with the changes in the FTIR data. With the well-established method for ion exchange, the current data corroborate that the majority of the anions were exchanged. We compiled the FTIR data of the individual polymers (prior to and after anion exchange) and compared them with the FTIR spectra of the salts used for the anion exchange to better clarify the changes in the spectra. Here, the distinctive -BF4 absorption band at 1013 cm-1 is clearly visible and can be directly related to the anion exchange (Figure 2-5).  

Comment 4: While the preliminary evaluation of I4-BF₄ as a liner material is valuable, a more direct comparison with commercial materials would strengthen the application argument.

Response 4: As suggested, we included the mechanical data of commercial polymers currently used as liner as a benchmark and added a paragraph in the discussion section (page 10).

Comment 5: Define all abbreviations upon first use, e.g., pTSA.

Response 5: Has been corrected accordingly.

Comment 6: In the supplementary materials, the 13C NMR spectra should be provided.

Response 6: 13C NMR experiments of M1 and M2 have been included. For the polymers, we did not perform 13C NMR experiments as the structures of the ionenes were well assigned by 1H NMR data.

Comment 7: page 6, line 220, boride should read bromide.

Response 7: Has been corrected accordingly.

Comment 8: References 50 and 51 miss specific chapter authors or page numbers.

Response 8: Thank you for pointing out the incorrect references. Reference 50 and 51 are the same reference and the mistake was corrected and the missing editor names and volume number were included (Reference 55 in revised manuscript).

Round 2

Reviewer 2 Report

Comments and Suggestions for Authors

The author made revisions based on the reviewer's suggestions, and therefore, the reviewer agreed to accept the paper.